# PhenGenVar: A User-Friendly Genetic Variant Detection and Visualization Tool for Precision Medicine

**DOI:** 10.3390/jpm12060959

**Published:** 2022-06-12

**Authors:** JaeMoon Shin, Junbeom Jeon, Dawoon Jung, Kiyong Kim, Yun Joong Kim, Dong-Hoon Jeong, JeeHee Yoon

**Affiliations:** 1Department of Computer Engineering, Hallym University, Chuncheon 24252, Korea; shin@dbcls.rois.ac.jp (J.S.); jjbblue12@hallym.ac.kr (J.J.); kozer@naver.com (D.J.); 2Database Center for Life Science, Joint Support-Center for Data Science Research, Research Organization of Information and Systems, Chiba-Ken, Kashiwa-Shi 277-0971, Japan; 3Department of Electronic Engineering, Kyonggi University, Suwon 16227, Korea; eye4eye@kyonggi.ac.kr; 4Department of Neurology, Yonsei University College of Medicine, Seoul 03722, Korea; yunjkim@yuhs.ac; 5Department of Neurology, Yongin Severance Hospital, Yonsei University Health System, Yongin 16995, Korea; 6Department of Life Science and Multidisciplinary Genome Institute, Hallym University, Chuncheon 24252, Korea

**Keywords:** precision medicine, NGS, exome browser, genetic variations

## Abstract

Precision medicine has been revolutionized by the advent of high-throughput next-generation sequencing (NGS) technology and development of various bioinformatic analysis tools for large-scale NGS big data. At the population level, biomedical studies have identified human diseases and phenotype-associated genetic variations using NGS technology, such as whole-genome sequencing, exome sequencing, and gene panel sequencing. Furthermore, patients’ genetic variations related to a specific phenotype can also be identified by analyzing their genomic information. These breakthroughs paved the way for the clinical diagnosis and precise treatment of patients’ diseases. Although many bioinformatics tools have been developed to analyze the genetic variations from the individual patient’s NGS data, it is still challenging to develop user-friendly programs for clinical physicians who do not have bioinformatics programing skills to diagnose a patient’s disease using the genomic data. In response to this demand, we developed a Phenotype to Genotype Variation program (PhenGenVar), which is a user-friendly interface for monitoring the variations in a gene of interest for molecular diagnosis. This allows for flexible filtering and browsing of variants of the disease and phenotype-associated genes. To test this program, we analyzed the whole-genome sequencing data of an anonymous person from the 1000 human genome project data. As a result, we were able to identify several genomic variations, including single-nucleotide polymorphism, insertions, and deletions in specific gene regions. Therefore, PhenGenVar can be used to diagnose a patient’s disease. PhenGenVar is freely accessible and is available at our website.

## 1. Introduction

Precision medicine, also known as personalized medicine, is a new emerging field of medicine that combines the individual patient’s clinical phenotypes, health records, and diverse omics data for tailored diagnosis, treatment, and prevention of human diseases [1]. With the completion of the human genome project and advances in next-generation DNA sequencing (NGS) technologies, it is now possible to identify the patient’s genetic variations at single-nucleotide resolution and interpret the phenotypic consequences of them in human diseases [2,3,4,5].

Several NGS approaches have been used to identify complex human disease-associated genetic variations. These include whole-genome sequencing, whole-exome sequencing, and targeted gene panel sequencing [6,7]. Recent large-scale consortia efforts have facilitated the meta-analysis of genome-wide association studies (GWAS) to identify genetic variants associated with human diseases, such as Alzheimer’s disease, cancers, coronary artery disease, and type 2 diabetes mellitus [4,8,9,10]. For instance, several genes, such as *APOE*, *CD33*, *CLU*, *CR1*, *EPHA1*, *PICALM*, and *TREM2*, have been identified as major genetic risk factors for Alzheimer’s disease [11,12,13,14,15]. *BRCA1* and *BRCA2* mutations account for approximately 5% of all breast cancer cases [16,17]. However, pathogenic mutations in *BRCA1* and *BRCA2* increase the risk of breast cancer by 65% and 45%, respectively [18]. This implies that despite the low total heritability explained by polymorphisms in these genes, monitoring genetic variations in *BRCA1* and *BRCA2* is advantageous for the diagnosis of breast cancer in high-risk patients [19]. Therefore, it is essential to obtain a catalog of genes associated with specific human diseases and phenotypes in order to facilitate genetic risk prediction. There are many clinical databases, such as Medical Subject Headings (MeSH), the National Cancer Institute’s (NCI) Thesaurus, Online Mendelian inheritance in man (OMIN), SNOMEDCT, United Medical Language System (UMLS), and Human Phenotype Ontology (HPO) [20,21,22,23,24,25]. Among these, HPO provides a more comprehensive resource that contains semantic links for disease genes and ontologies for computational analysis of human phenotypes.

NGS data of individual patients contain diverse types of genetic variations, such as single nucleotide variants (SNVs), insertions and deletions (indels), and genetic aberrations, such as inversions and translocations. There are many computational tools for variant calling from NGS data, such as DeepVariant, DELLY, FermiKit, GATK HaplotyperCaller (GATK HC), Pindel, Platypus, Strelka2, and VarScan [26,27,28,29,30,31,32,33]. Using these variant detection tools, it is possible to detect the genetic variants in the disease-related genes for molecular diagnosis. The genetic variants identified from the NGS data are stored as variant call format (VCF), mutation annotation file (MAF), mutation file format (MUT), or other types of files. Among these, VCF is the most widely used community standard for storing mutation data [34]; however, the analysis of VCF files requires intensive bioinformatic expertise. Therefore, there is a need to develop tools for managing and visualizing VCF data. The currently available VCF processing programs include SNVerGUI, database.bio, DaMold, mirVAFC, GAVIN, and gNOME [35,36,37,38,39,40]. In addition, other tools, such as integrative genomic viewer (IGV), variant call miner (VCF-Miner), VCF.Filter, myVCF, BrowseVCF, and VCF-Server, were also developed to identify disease-associated genetic variants from NGS data [41,42,43,44,45,46]. Among these, IGV is the most widely used tool for visualizing genome variation data. However, there is still a need to develop an easy and comprehensive tool for non-bioinformatician researchers to analyze a patient’s genomic data for diagnosis and clinical decision support.

In this study, we developed the Phenotype to Genotype Variation (PhenGenVar) browser application as an intuitive user interface. This tool was designed for physicians or researchers to monitor the patient’s genetic variations from the set of selected genes that are associated with a specific disease or phenotype. With exome and genome browsers in a PhenGenVar application, it is possible to detect genetic variations at the gene or exon level, as well as at the single-nucleotide level.

## 2. Materials and Methods

### 2.1. Development of Visual Interface for PhenGenVar Browser

The PhenGenVar browser application was written in the C# programing language using Microsoft.Net framework. The application was developed as a graphic interface that can be performed on a Windows operating system. The human genome reference dataset, including the position, annotation, and amino acid composition, was obtained from the UCSC genome browser database [47]. To remove duplicated reads and minimize errors in sequence alignment, the flag and concise idiosyncratic gapped alignment report (CIGAR) string fields were utilized for error correction and realignment of the read data in the BAM file [48]. We adopted a data partitioning and indexing method for efficient visualization of large-scale genomic data. Genomic data are divided into partitions that can be managed and accessed separately, using an efficient indexing scheme. To reduce the information loss caused by partitioning, each partitioned region included overlapping regions between the flanking partitions. In display processing, indexed partitions are loaded into memory, and the corresponding genomic regions are displayed on the screen according to the resolution. In the browser, the resolution of the sequence alignment was adjusted from a 100 bp to 100 kb window. To minimize the decline in speed to render many output images of multiresolution, Direct2D-based rendering was used.

### 2.2. Database Embedded in the PhenGenVar Application

Human reference genome data were downloaded and pre-installed in the PhenGenVar application as the hg19 and hg38 versions [47]. The SNP database (dbSNP) was downloaded from the National Center for Biotechnology Information [49]. The dbSNP versions available for the hg19 human genome are SNP138, SNP141, SNP142, SNP144, SNP146, SNP147, SNP150, and SNP151. The dbSNP versions available for hg38 are SNP141, SNP142, SNP144, SNP146, SNP147, SNP150, and SNP151. The gene sets were downloaded from the human phenotype ontology (HPO) to provide a list of genes associated with human disease and phenotype [24].

### 2.3. Sample Data Used in This Study

To test the PhenGenVar application for a patient’s diagnosis, publicly available personal whole-genome sequence data were retrieved from the 1000 Genomes Project with sample accession number NA11995 [50].

### 2.4. Implementation

The PhenGenVar is available at http://dblab.hallym.ac.kr/PhenGenVar, and the copyright of the software was provided freely. PhenGenVar was implemented on a Windows 10 platform with an Intel Core i7 3.3GHz CPU, 32 GB main memory, and 1TB hard drive.

## 3. Results

### 3.1. Development of a PhenGenVar Browser Application

We developed a PhenGenVar browser application as an intuitive user interface to enable clinical physicians and biological researchers to explore genetic variations in a gene of interest for molecular diagnosis. It allows users to browse for genomic regions and variants associated with a specific phenotype. To protect the personal information of patients and enhance the speed of the analysis, the application was developed as a personal computer-based software program that runs locally instead of a web-based program.

PhenGenVar consists of two separate browsers: an exome browser and a genome browser (Figure 1). The exome browser of PhenGenVar is the main program designed for gene-level analysis by selecting a user-defined phenotype-related gene set and then monitoring the genetic variations of the specific gene. The exome browser output displayed the corresponding genomic regions with the reported variants and detailed information. The genome browser can subsequently be called from the exome browser and utilized to closely browse the variant calling results based on read alignment. Thus, the PhenGenVar application provides a convenient and intuitive user interface at various levels of genomic resolution.

### 3.2. PhenGenVar Exome Browser for Gene-Level Variation Analysis

The exome browser of PhenGenVar is an overview page for gene-level analysis of genetic variations from a patient’s genomic data, such as exome sequencing or whole-genome sequencing. This browser page mainly consists of data uploading areas, gene and variant filtering panels, and an exon viewer panel (Figure 1A).

To begin data analysis, it is necessary to upload the binary sequence alignment map (BAM) file and VCF file, both of which are generated from the patient’s exome or whole-genome sequencing data. To explore the genetic variations of genes related to the patient’s disease phenotype, a gene list can be created or typed individually as HUGO Gene Nomenclature Committee (HNGC) official gene symbol names [51]. In addition, the exome browser provides the gene groups registered in human phenotype ontology (HPO), which is a comprehensive resource for over 13,000 terms related to phenotypic abnormalities found in human diseases [24]. To analyze the genetic variations of each gene using the VCF/BAM file data, users can flexibly select a specific human reference genome sequence version and dbSNP version.

Once the patient’s genome data and gene groups are uploaded, the analyzed output data are represented as a summary table, as shown in Figure 2A. This table shows the SNP data for each gene in the selected gene group. The SNP data include not only the SNPs registered in the dbSNP database [52] but also novel SNPs, which might be rare variations found in the patient’s genome. A set of genes with genetic variation can be selected using the gene filter panel of the exome browser, as shown in Figure 2B. In this filter panel, allele frequency values can also be adjusted to narrow down genes with more significant genetic variations. In addition to gene filtering, a more detailed filtering option is available using the VCF filter panel (Figure 2C, upper panel). Once a specific gene is selected from the gene filter panel, all variant information of the corresponding gene is represented in the exon variant call panel of the exome browser (Figure 2C, lower panel). This panel provides detailed variant information, such as position, dbSNP reference number (rsID), sequence alterations compared to the reference genome, and type of variations. In the above VCF filter panel, additional filtering was performed by clicking the type of sequence variations and adjusting the allele frequency.

The final output of the exome browser is shown in the main exon view panel, in which the gene structure, variants along the reference sequence, and read alignment are represented (Figure 3). To test how the PhenGenVar application can be utilized to detect genetic variation, we uploaded a publicly available personal whole-genome sequence data from 1000 human genome project and monitored it to identify various genetic variations [50]. In the gene structure area, exons with or without sequence variations are represented in blue or green boxes, respectively. The read alignment viewer showed various types of sequence variations, such as SNPs (Figure 3A), deletions (Figure 3B), and insertions (Figure 3C). Therefore, we were able to identify specific sequence variations in the read alignment viewer. The sequence variation regions can also be represented by selecting the specific variation from the exon variant call panel (Figure 3C). Since the PhenGenVar application was designed to show the read alignment in the 30 bp upstream and downstream of the exon as a default setting, the exome browser also provides sequence variation in the intron. The scope of intron area output was adjustable in the Viewer Control Panel. We demonstrated that the exome browser of the PhenGenVar application can be used to identify genetic variations, which might be related to the patient’s disease phenotype.

### 3.3. PhenGenVar Genome Browser for Single Base-Resolution Analysis

The PhenGenVar genome browser was designed to present detailed read alignment information at single-nucleotide resolution (Figure 1B). The genome browser is popped up from the exome browser by selecting variants from the exon variant call panel or by double clicking the variant region from the main view panel. In addition, read alignment can be performed in a specific region of the genome by typing the chromosome position or gene name in the control panel of the genome browser. The resolution of the read alignment panel can be adjusted by changing the levels in the trackbar of the control panel from 59 bp to 121,704 bp. Thus, the structure of the corresponding gene and its neighboring genes can be shown at low resolution, while the nucleotide sequences and coded amino acid sequences can be shown at higher resolution.

To test whether the PhenGenVar genome browser can be utilized to identify specific genetic variations at single-nucleotide resolution, we monitored the various types of sequence variations using the personal genome sequencing data that were used above. As shown in Figure 4, we identified SNP variant, deletion, and insertion in specific genes of personal genome. The single nucleotide change from C to T in an *ATR* gene, in which the synonymous mutation does not change the coding amino acid, was identified from all the sequence reads (Figure 4A). In an *MST1L* gene, deletion of five nucleotide was confirmed in half of the sequence reads (Figure 4B). This implies that one of two paired chromosomes harbors a mutated MST1L gene with a frameshift. We also identified an insertion of additional CTC sequence in a *PRDM2* gene (Figure 4C). Since about half of the sequence reads contained this insertion, we estimate that one of two paired chromosomes has a mutated *PRDM2* gene with an additional CTC insertion. As a result, we showed that the visual inspection with a PhenGenVar genome browser is an effective and powerful tool for variant call validation, reducing the number of false positives and assisting in the confirmation of true genetic variant discoveries.

## 4. Discussion

In this study, we developed a knowledge-based PhenGenVar browser for disease-gene variation visualization for clinical diagnosis. Using personal whole-genome sequencing data, we proved that the PhenGenVar browser is useful for identifying genetic variations, such as SNPs, insertions, and deletions of specific genes from the user-defined gene set. Compared to other genome browsers, such as IGV, which were designed to monitor genetic variations of all genomic regions from large-scale datasets, our program was specialized to quickly detect the genetic variations of genes of interest from the patient’s exome or whole-genome sequencing data. Moreover, because the PhenGenVar can be installed on a personal computer, the security of the patient’s personal information can be better guaranteed.

The advent of high-throughput NGS technology has accelerated the discovery of common and rare disease-associated genetic variants in large population genomic data [53,54]. Based on this information, personal exome or whole-genome sequencing has been widely used to identify genetic variations and their associations with various human diseases [54,55]. For instance, *BRCA1* and *BRCA2* are partially responsible for breast and ovarian cancers, and their SNVs are widely used for cancer diagnosis [56]. Since our knowledge of the genes related to a specific disease phenotype has greatly increased, it is necessary to diagnose a patient’s disease using a precise and fast method. The PhenGenVar application can be used as a tool for this purpose. Compared to previous tools, one advantage of the PhenGenVar program is that the exome browser is specifically designed to quickly monitor the genetic variations in each exon and its exon–intron junction area, which is critical for the deleterious mutations of disease-responsible genes. To prove this, we showed that the application could easily detect genetic variations in specific genes using randomly selected whole-genome sequencing data from an anonymous person. Thus, we believe that the PhenGenVar can be widely used for clinical diagnosis utilizing the relatively cost-effective exome sequencing data. Moreover, this method can be applied to custom-targeted gene panel sequencing data [57].

The main objective of PhenGenVar application is to quickly identify genetic variations from a knowledge-based gene set. Thus, the precise diagnosis using this application relies on the quality of the gene set associated with a specific disease. The gene sets from the HPO were pre-installed in the current version of PhenGenVar. We will update more disease gene sets upon user request in the next version.

## 5. Conclusions

PhenGenVar is a freely available program developed with the aim of supporting clinical diagnosis using the patient’s NGS data. It is designed to monitor the genetic variations of selected gene lists that are associated with specific disease and phenotype. This user-friendly program is developed for researchers or physicians to analyze the patient’s genetic data with no programing knowledge requirement. Because the PhenGenVar browsers provide the comprehensive GUI-based VCF visualization, it allows users to manage, filter, query, and export the variant results in a fast and effective way (Appendix A). We expect that this tool can be widely used by the medical physicians to diagnose the patient’s diseases from their genomic data.

## Figures and Tables

**Figure 1 jpm-12-00959-f001:**
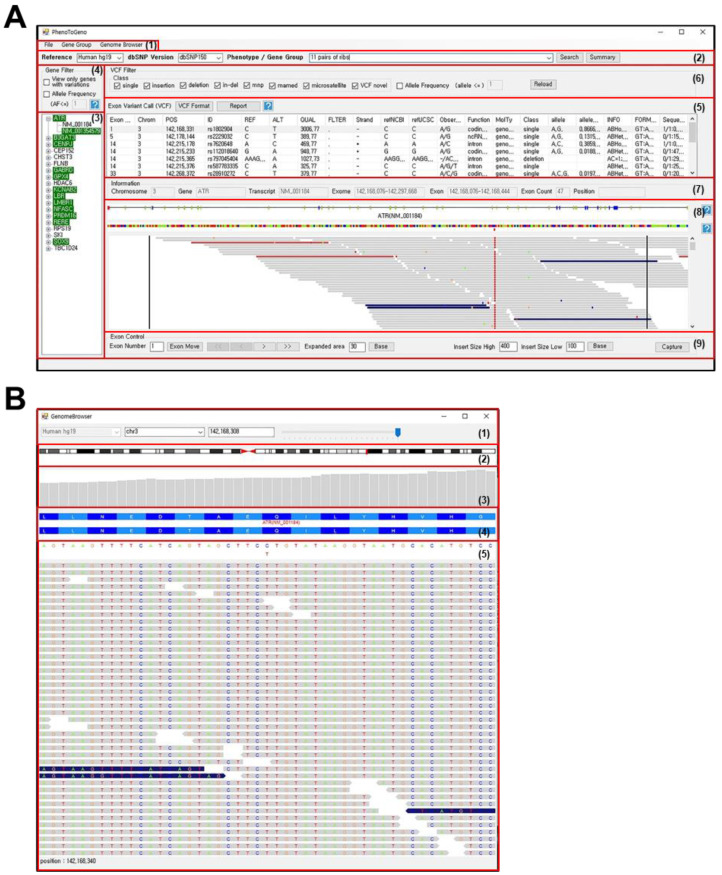
Main pages of PhenGenVar exome and genome browser. (**A**) Main page of PhenGenVar exome browser. The browser consists of menu bar (1), control panel (2), gene/transcript view panel (3), gene filter panel (4), exon variant call panel (5), VCF filter panel (6), information panel (7), main exon view panel (8), and exon view control panel (9). (**B**) Main page of PhenGenVar genome browser. The browser consists of control panel (1), cytoband panel (2), coverage graph panel (3), genetic structure panel (4), and main genome view panel (5).

**Figure 2 jpm-12-00959-f002:**
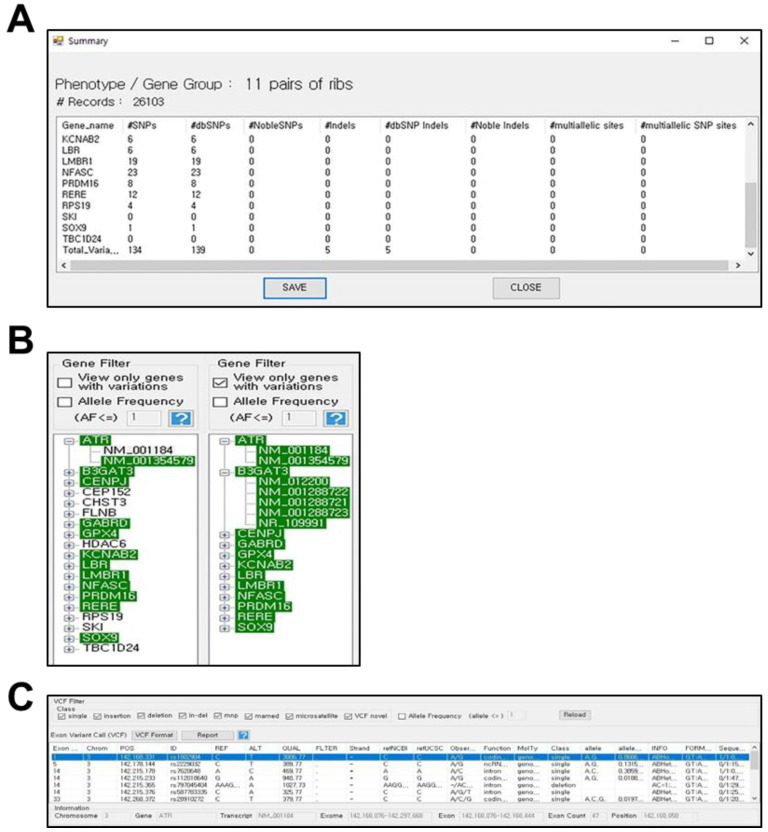
Output data of the exon variant information. (**A**) Statistical summary of the variant information from the VCF file. (**B**) Example of the gene filter panel and gene/transcript view panel. The genes and transcript with variants are highlighted with green color. (**C**) Example of the variant output in the VCF filter panel and the exon variant call panel.

**Figure 3 jpm-12-00959-f003:**
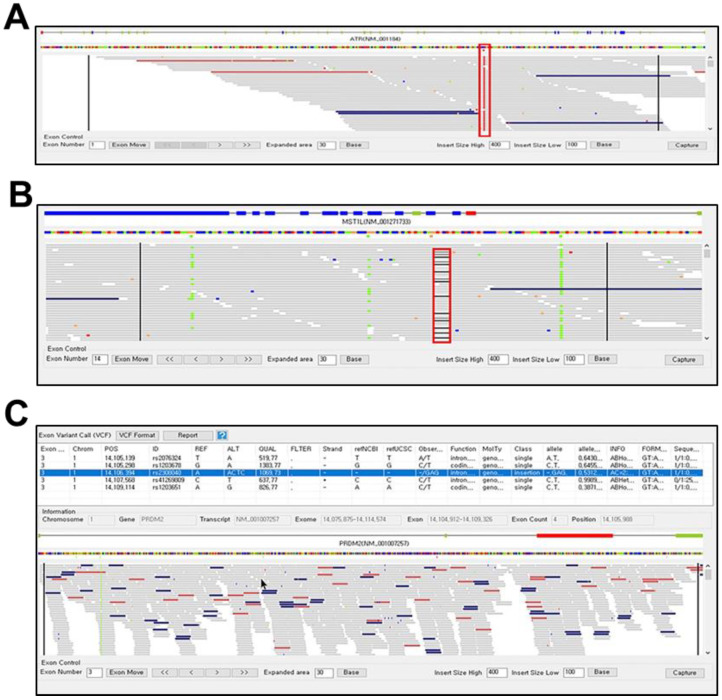
Identification of genetic variants in an exome browser. (**A**) Single nucleotide polymorphism (SNP) in an ATR gene. The red box indicates the position of SNP. (**B**) Deletion in a MST1L gene. The deleted area is shown in a red box. (**C**) Insertion of the sequences in a PRDM2 gene. The position of the insertion is indicated by the arrow.

**Figure 4 jpm-12-00959-f004:**
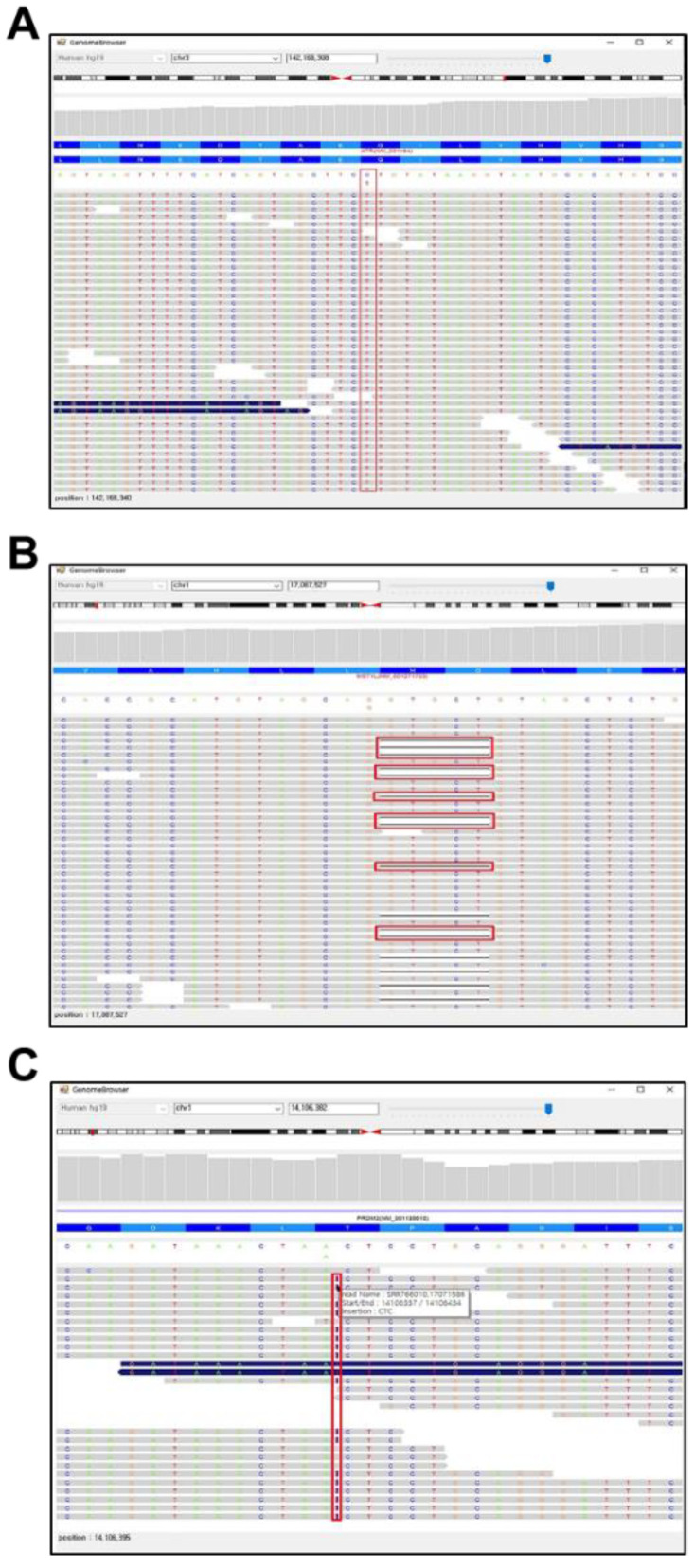
Identification of genetic variants in a genome browser with a single-nucleotide resolution. (**A**) Single nucleotide polymorphism (SNP) in an *ATR* gene. (**B**) Deletion in a *MST1L* gene. (**C**) Insertion of the sequences in a *PRDM2* gene. The positions of genetic variants are indicated with red boxes.

## Data Availability

All data presented in this study are available on request from the corresponding author. PhenGenVar is freely accessible and is available at http://dblab.hallym.ac.kr/PhenGenVar.

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
