# Peer review of "PhenGenVar: A User-Friendly Genetic Variant Detection and Visualization Tool for Precision Medicine"

_jpm, 2022, doi:10.3390/jpm12060959_

Round 1

Reviewer 1 Report

The authors have developed PhenGenVar, a visualization and analysis platform for precision medicine. It requires BAM, VCF file and gene list as inputs and present options to visualize the variants at different nucleotide resolution level. The rapid analysis can be due to the specified focus to certain user-defined genes and software implementation. The manuscript includes detailed manual of the tool and should be intuitive to use. However, a major concern is lack of explicit comparison with existing tool IGV by Broad Institute. The paper does make claim of quicker analysis but does not provide any comparison. Nonetheless, since it is freely available an alternative is available to the precision medicine community.

A couple of suggestions to increase the readability of the manuscript.

1. What do the authors mean by "phenotypic consequences of human diseases" in line 48? 

2. In figure 1, the panels listed in the figure caption can be marked or highlighted in the figure as well.  

Author Response

The authors have developed PhenGenVar, a visualization and analysis platform for precision medicine. It requires BAM, VCF file and gene list as inputs and present options to visualize the variants at different nucleotide resolution level. The rapid analysis can be due to the specified focus to certain user-defined genes and software implementation. The manuscript includes detailed manual of the tool and should be intuitive to use. However, a major concern is lack of explicit comparison with existing tool IGV by Broad Institute. The paper does make claim of quicker analysis but does not provide any comparison. Nonetheless, since it is freely available an alternative is available to the precision medicine community.

Answer: We appreciate the reviewer’s comments. To address the concern about comparison with other programs, such as IGV, we described the comparison in the discussion part on lines 250-256 and line 265-268.

A couple of suggestions to increase the readability of the manuscript.

1. What do the authors mean by "phenotypic consequences of human diseases" in line 48? 

Answer: Thank you for pointing out the unclearness of the sentence. Therefore, we rewrote the sentence as follows:

“With the completion of the human genome project and advances in next-generation DNA sequencing (NGS) technologies, it is now possible to identify the patient’s genetic variations at single-nucleotide resolution and interpret the phenotypic consequences of them in human diseases.”

2. In figure 1, the panels listed in the figure caption can be marked or highlighted in the figure as well.  

Answer: We revised the Figure 1 as suggested.

Reviewer 2 Report

In this work, the authors have developed a user-friendly program “Phenotype to Genotype Variation program (PhenGenVar)” for non-bioinformatician researchers to analyse a patient’s genomic data for diagnosis and clinical decision support. The robustness of the developed program has been tested by analysing whole-genome sequencing data of an anonymous person from the 1000 human genome project data.

PhenGenVar consists of two separate browsers. The exome browser has been designed for gene-level analysis by selecting a user-defined phenotype-related gene set and then monitoring the genetic variations of the specific gene. While, the genome browser is utilised to closely browse the variant calling results based on read alignment, that can be called from the exome browser. Compared to other available tools, their program has been specifically designed to quickly monitor the genetic variations in each exon and its exon-intron junction area, which is critical for the deleterious mutations of disease-responsible genes.

The work presented by the authors is interesting and will aid Clinicians to diagnose the patient’s disease from the genomic data of the patient.

Comments:

1)     The authors should add a flow chart, highlighting the key steps of the developed program.

2)     It will be useful to provide a brief comparison on the performance of the developed program with the known tools.

Author Response

In this work, the authors have developed a user-friendly program “Phenotype to Genotype Variation program (PhenGenVar)” for non-bioinformatician researchers to analyse a patient’s genomic data for diagnosis and clinical decision support. The robustness of the developed program has been tested by analysing whole-genome sequencing data of an anonymous person from the 1000 human genome project data.

PhenGenVar consists of two separate browsers. The exome browser has been designed for gene-level analysis by selecting a user-defined phenotype-related gene set and then monitoring the genetic variations of the specific gene. While, the genome browser is utilised to closely browse the variant calling results based on read alignment, that can be called from the exome browser. Compared to other available tools, their program has been specifically designed to quickly monitor the genetic variations in each exon and its exon-intron junction area, which is critical for the deleterious mutations of disease-responsible genes.

The work presented by the authors is interesting and will aid Clinicians to diagnose the patient’s disease from the genomic data of the patient.

Comments:

1) The authors should add a flow chart, highlighting the key steps of the developed program.

Answer: We appreciate the reviewer’s comments. In the revised manuscript, we added a supplementary Figure S1 to introduce the simple steps for using the PhenGenVar Browser.

2) It will be useful to provide a brief comparison on the performance of the developed program with the known tools.

Answer: To address the reviewer’s comment, we described the comparison in the discussion part on lines 250-256 and line 265-268.